# Chromogranin A in the Laboratory Diagnosis of Pheochromocytoma and Paraganglioma

**DOI:** 10.3390/cancers11040586

**Published:** 2019-04-25

**Authors:** Radovan Bílek, Petr Vlček, Libor Šafařík, David Michalský, Květoslav Novák, Jaroslava Dušková, Eliška Václavíková, Jiří Widimský, Tomáš Zelinka

**Affiliations:** 1Institute of Endocrinology, Národní 8, 116 94 Prague, Czech Republic; evaclavikova@endo.cz; 2Department of Nuclear Medicine and Endocrinology, Motol Teaching Hospital and Second Faculty of Medicine, Charles University, V Úvalu 84, 15006 Prague 5, Czech Republic; petr.vlcek@fnmotol.cz; 3Urology Clinic, V Pražské bráně 74, 26601 Beroun, Czech Republic; lsafarik@centrum.cz; 41st Department of Surgery-Department of Abdominal, Thoracic Surgery and Traumatology, First Faculty of Medicine, Charles University and General University Hospital, U Nemocnice 2, 12808 Prague 2, Czech Republic; david.michalsky@lf1.cuni.cz; 5Department of Urology, First Faculty of Medicine, Charles University and General University Hospital, Ke Karlovu 4, 12808 Prague 2, Czech Republic; kvetoslav.novak@vfn.cz; 6Institute of Pathology, First Faculty of Medicine, Charles University in Prague and General University Hospital, Studničkova 2, 12800 Prague 2, Czech Republic; jaroslava.duskova@lf1.cuni.cz; 7Center for Hypertension, 3rd Medical Department-Department of Endocrinology and Metabolism, First Faculty of Medicine, Charles University and General University Hospital, U Nemocnice 1, 12808 Prague 2, Czech Republic; jwidi@lf1.cuni.cz (J.W.J.); Tomas.Zelinka@lf1.cuni.cz (T.Z.)

**Keywords:** chromogranin A, metanephrines, pheochromocytoma, paraganglioma

## Abstract

This work discusses the clinical performance of chromogranin A (CGA), a commonly measured marker in neuroendocrine neoplasms, for the diagnosis of pheochromocytoma/paraganglioma (PPGL). Plasma CGA (cut-off value 150 µg/L) was determined by an immunoradiometric assay. Free metanephrine (cut-off value 100 ng/L) and normetanephrine (cut-off value 170 ng/L) were determined by radioimmunoassay. Blood samples were collected from PPGL patients preoperatively, one week, six months, one year and two years after adrenal gland surgery. The control patients not diagnosed with PPGL suffered from adrenal problems or from MEN2 and thyroid carcinoma. The clinical sensitivity in the PPGL group of patients (n = 71) based on CGA is 90% and is below the clinical sensitivity determined by metanephrines (97%). The clinical specificity based on all plasma CGA values after surgery (n = 98) is 99% and is the same for metanephrines assays. The clinical specificity of CGA in the control group (n = 85) was 92% or 99% using metanephrines tests. We can conclude that plasma CGA can serve as an appropriate complement to metanephrines assays in laboratory diagnosis of PPGL patients. CGA is elevated in PPGLs, as well as in other neuroendocrine or non-neuroendocrine neoplasia and under clinical conditions increasing adrenergic activity.

## 1. Introduction

In this work we present our experiences with radioimmunoassay of plasma chromogranin A (CGA) in the laboratory diagnosis of neuroendocrine tumors classified as pheochromocytoma (PCC) and paraganglioma (PGL). Radioimmunoassay of plasma methanephrines were also performed.

Neuroendocrine cells are widely dispersed cells with dense core granules similar to those dense core granules present in serotonergic neurons (neuro properties), which store bioactive amines and peptide hormones (endocrine properties) [1,2]. These cells do not contain synapses [1,3]. Neuroendocrine neoplasms are a heterogeneous group of tumors, including malignancies from several anatomic areas that also include pheochromocytomas and paragangliomas, commonly denoted PPGLs. Neuroendocrine neoplasms are classified into well-differentiated neuroendocrine tumors, poorly differentiated neuroendocrine carcinomas, and mixed adenoneuroendocrine carcinomas showing more than 30% of neuroendocrine cells and additional components [4,5]. Neuroendocrine neoplasms can be divided into functional and non-functional tumors; the former ones are usually diagnosed at an earlier stage due to endocrine symptoms related to hormonal production, whereas the non-functional tumors remain silent and are frequently diagnosed when metastasis has already occurred [2,5]. Neuroendocrine neoplastic cells have been described in the central nervous system, respiratory tract, the larynx, gastrointestinal tract, thyroid, skin, urogenital system, breast and lung [2]. Well differentiated neuroendocrine tumor cells synthesize, store and secrete chromogranin A (CGA) and amines; metastatic neuroendocrine carcinomas have fewer cytoplasmic secretory granules [1]. Serum CGA is a commonly measured marker in neuroendocrine neoplasms [3]. 

PPGLs are rare neuroendocrine tumors of a chromaffin cell origin usually found in the adrenal medulla and other ganglia of the nervous system [6]. PCC is a tumor arising from adrenomedullary chromaffin cells that commonly produces catecholamines. PGL is a tumor derived from extra-adrenal chromaffin cells of the sympathetic paravertebral ganglia (thorax, abdomen, pelvis) producing catecholamines or parasympathetic ganglia located in the neck and at the base of the skull, which is often non-secretory [7,8,9]. PGLs were identified in the head and neck, being most frequent in the carotid body, followed by jugulotympanic paraganglia, vagal nerve and ganglion nodosum, as well as laryngeal paraganglia. Abdominal sites include urinary bladder tumors; other unusual sites are peri-adrenal, para-aortic, inter-aortocaval, and paracaval retroperitoneal sites, as well as tumors in the thyroid, parathyroid, pituitary, gut, pancreas, liver, mesentery, lung, heart and mediastinum [10]. The adrenal tumor localization is usually found in 80–85% of patients, and extra-adrenal sympathetic and parasympathetic PGL are observed in 10–20% of patients [11,12]. While most PPGLs are benign, approximately 10% of the PCCs and 20–25% of the PGLs are malignant [8,13,14]. 

PPGLs have considerable genetic heterogeneity [15] associated with various clusters based to different patient outcomes and underlying genetics. The *pseudohypoxia group* is characterized by somatic or germline mutations and silent or dopaminergic and/or noradrenergic secretory profiles in the tricarboxylic acid cycle related to succinate dehydrogenase subunits *SDHA*, *SDHB*, *SDHC*, *SDHD*, *SDHAF1*, *SDHAF2* (together *SDHx*) or in fumarate hydratase (*FH*), a second enzyme in the tricarboxylic acid cycle, in von Hippel-Lindau disease (*VHL*), endothelial PAS domain 1 (*EPAS1*0, also known as hypoxia–inducible factor 2α (*HIF2A*)), and prolylhydroxylases *PHD1* and *PHD2*. The *wnt signaling group* includes somatic mutations in cold shock domain containing E1 (*CSDE1*), α thalassemia/mental retardation syndrome X-linked (*ATRX*) and mastermind-like transcriptional coactivator (3*MAML3*) with mixed noradrenergic and adrenergic secretory phenotype. The *kinase signaling group* consists of germline or somatic mutations in *RET* proto-oncogene (syndrome MEN 2A, 2B), neurofibromin 1 (*NF1*), transmembrane protein 127 (*TMEM127*), MYC-associated factor X (*MAX*), kinesin-like protein (*KIF1BB*), receptor tyrosine kinase (*MET*) and GTPase, Harvey rat sarcoma viral oncogene homolog (*HRAS*) with adrenergic or mixed noradrenergic and adrenergic secretory profiles [10,15]. Mutations in the mitochondrial succinate dehydrogenase enzyme complex subunit B (*SDHB*) and the tumor suppressor gene Von Hippel-Lindau (*VHL*) are mainly associated with malignancy [13]. 

Elevated levels of circulating CGA have been associated with almost all types of neuroendocrine neoplasms including PPGLs [5]. CGA belongs to the family of secretory chromogranin and secretogranin proteins (CGA, chromogranin B, chromogranin C or secretogranin II, secretogranins III, IV, V, VI, VII and VIII). These proteins are the driving force for the biogenesis of secretory chromaffin granules present in the diffuse neuroendocrine system [16]. CGA is an acidic hydrophilic glycoprotein abundantly expressed in large dense core vesicles of neuroendocrine cells [17]. CGA comprises at least 40% of the soluble proteins of the adrenal chromaffin granules [18]. Human CGA is encoded by the *CHGA* gene, located in chromosome 14q32.12 with eight exons and seven introns. It is transcribed and translated into a 439 amino acids protein with a molecular weight of 48 kDa which is co-stored and co-released with catecholamines [5,16]. The N-terminal domain of CGA is responsible for directing CGA into the secretory granules [19], and for binding to secretogranin III, the receptor for CGA requiring the presence of Ca^2+^ [4]. The CGA structure is described in Uniprot/SWISS-PROT database under the accession number P10645 and it consists of 18 amino acids (aa) long signal peptide (CGA 1–18) and 439 aa long CGA (together 457 aa). It includes multiple dibasic cleavage sites [5]. CGA is processed to a lesser extent within the secretory granules to yield bioactive peptides [20]. These peptide hormones such as vasostatin-1 (CGA 19–94), vasostatin-2 (CGA 19–131), pancreastatin (CGA 272–319), catestatin (CGA 370–390), parastatin (CGA 347–419), serpinin (CGA 429–454), chromofugin (CGA 47–66), chromostatin (CGA 124–143), chromactin I (CGA 173–194), chromactin II (CGA 195–221) or WE14 (CGA 316–329) have different biological functions. Generally the peptide hormones negatively modulate the neuroendocrine function [4,5] and are involved in regulation of the cardiovascular system, metabolism, innate immunity, angiogenesis and tissue repair [21].

The main biological role of CGA is to regulate calcium-mediated exocytosis [22]. The granin family has the capacity to bind calcium ions and the ability to form aggregates [5]. They are involved in vesicle sorting, in the generation of bioactive peptides and in the accumulation of soluble species such as catecholamines and Ca^2+^ at low pH to large dense core vesicles. CGA is synthesized in the rough endoplasmic reticulum, transported to the Golgi complex and packaged together with other secretory proteins/peptides and amines into immature granules, where it may be cleaved into the various derived peptides by specific processing enzymes. Upon acidification, secretory granules mature, and are ready for stimulation–induced release. Intact CGA controls the dense core granule biogenesis as well as the sorting and secretion of other bioactive molecules, and participates in the regulation of cytosolic calcium stores and granule exocytosis [5,23]. The pH gradient across the membrane of large dense core vesicles is responsible for maintaining the high concentrations of amines, Ca^2+^ and ATP inside the vesicles. The pH gradient depends on the activity of a vesicular H^+^-proton pump ATPase, which is continuously pumping H^+^ to acidify the vesicles [24]. Treatment of patients with proton pump inhibitors (PPIs) can increase the concentrations of CGA in circulation. 

CGA is an essential protein for PPGLs [25]. High levels of CGA, co-stored and co-secreted with catecholamines, may indicate tumor mass and malignancy in PPGL patients and can be used to monitor response and relapse [13]. Although non-specific for PPGL, CGA may facilitate diagnostic evaluation of e.g., SDHB-related PPGL, especially where the measurement of plasma metanephrines could otherwise be delayed by decreased availability or cost restriction [26]. High levels of CGA at the time of PPGL diagnosis were associated with the presence of metastases according to the histological evaluation of primary PPGL tumors with a PASS scoring scale, which is based on the histological evaluation of large nest or diffuse growth (>10% of tumor volume) (2 points), central or confluent necrosis (2 points), high cellularity (2 points), cellular monotony (2 points), tumor cell spindling (2 points), mitoses >3/10 HPF (2 points), atyp. mitoses (2 points), periadrenal fat infiltration (2 points), angioinvasion (1 point), transcapsular invasion (1 point), extreme pleomorphism (1 point) and nuclear hyperchromasia (1 point) [27]. The PASS score of PPGL patients usually exceeds 4 points [28]. A value of less than four out of 20 points indicates the benign character of a tumor [27]. The PASS score shows the possibility of malignancy, and its value higher than four indicates uncertain biological behavior. Increasing the score increases the likelihood of malignancy. However, the real evidence of malignancy is only the presence of metastasis in PPGL patients [7]. CGA is physiologically released via exocytosis by both functioning and non-functioning tumors [29]. A significant positive relationship was demonstrated between tumor mass and serum CGA levels [30,31,32]. Abnormally high circulating CGA levels are a typical feature of patients with neuroendocrine tumors and the detection of circulating CGA has a high sensitivity and specificity for the diagnosis of these tumors [29]. Plenty of misdiagnoses or delayed diagnoses still occur due to silent or weak clinical manifestation, especially for non-functioning neuroendocrine tumors [33]. CGA is a widely used biomarker for the assessment of neuroendocrine neoplasms, mainly of a gastroenteropancreatic origin [4]. It is elevated in approximately 90% of gut neuroendocrine tumors; the highest values are noted in ileal and gastrointestinal neuroendocrine tumors associated with MEN1. Pancreatic neuroendocrine tumors also have elevated values [1]. CGA was significantly higher in patients affected by hepatocellular carcinoma [34], gastroenteropancreatic neuroendocrine tumors [35], and in primary and metastatic small intestinal neuroendocrine tumors [36]. Gastric type I, pituitary and parathyroid tumors had lower values [1]. CGA is more frequently elevated in well-differentiated tumors compared to poorly differentiated neuroendocrine tumors [1]. 

Falsely elevated CGA levels are observed in patients treated with proton pump inhibitors (PPIs) or other acid-blocking medications [1,4,5,37]. This effect of the PPIs is fully eliminated after discontinuation of the PPI for 2 weeks [38]. Chronic renal insufficiency (CRI) can also substantially increase the concentration of circulated CGA [39] and may led to concentrations of CGA as high as those detected in patients with neuroendocrine neoplasm [40]. Other diseases of the alimentary tract affecting CGA concentration are chronic gastritis [41], chronic hepatitis, liver cirrhosis [42], pancreatitis, irritable bowel, and inflammatory bowel diseases [43]. Its concentration in circulation is also increased after myocardial infarction, acute coronary syndrome and heart failure [21,44,45]. The cardiovascular complications were observed in nearly 20% of patients with PPGLs due to an increased level of catecholamines [46]. 

We have presented in this study our experiences with the radioimmunoanalytical determination of plasma CGA in groups of patients who suffered from PPGLs. These data were partly published in the literature [47], and commentary on the data is given in the Discussion section. Patients with various endocrine disorders other than PPGLs were used as the control group.

## 2. Results

In all PPGL patients, CGA and methanephrines were determined by radioimmunoassay in EDTA plasma. The nature of the PPGL was differentiated by colored solid symbols as given in the legend of Figure 1. Plasma CGA results were not statistically dependent on either age or gender. 

As seen in Figure 1, clinical specificity and sensitivity was considerably influenced by administering proton pump inhibitors and in patients by chronic renal insufficiency. Recurrence of the disease (REC) was observed in two patients one week or two years after surgery. These results concerning PPI, CRI or REC were not included in the clinical specificity and sensitivity calculations listed in Table 1 and Table 2 since such increased values can be normalized by not administering PPI to patients, or long-term kidney failure or the recurrence of PPGL must be taken into consideration when interpreting the results as the case may be. Figure 1 describes CGA concentration in EDTA plasma of all samples of 78 PPGL patients and 86 controls in a given time period. Metanephrine (MN) and normetanephrine (NMN) using radioimmunoassay in EDTA plasma were also determined in these patients. Data from 71 PPGL patients were used for determination of sensitivity, the remaining seven patients were excluded from the study due to PPI, CRI, or REC (Table 1). One patient out of 86 control samples was also excluded from the same cause (CRI), so the calculation of specificity was done from a group of 85 patients.

PPGL patients are divided into PCC and PGL groups (total, males, females) with specified or not specified mutations in Table 1. Six of the PCC patients (none of the PGL patients) were metastatic (1 case of *RET* mutation, other mutations were not specified), two of these patients died prior to surgery. Nine PGL patients with preoperative CGA concentrations 355.9 ± 270.8 µg/L (Table 1) showed only noradrenergic phenotype with increased NMN and normal MN with one exception in which both MN and NMN were normal. Preoperative CGA concentrations 657.1 ± 492.7 µg/L (Table 1) were found in 62 PCC patients. According to Eisenhofer et al. [48], there was an adrenergic phenotype with elevated MN and variable NMN levels in 36 patients with a CGA concentration equal to 767.7 ± 513.7 µg/L. A noradrenergic phenotype with elevated NMN and normal or slightly increased MN to 120 μg/L was found in 25 patients with a CGA concentration of 516.9 ± 428.8 µg/L. Both MN and NMN were in normal reference ranges in one PCC patient. In general, the adrenergic phenotype of PPGL patients did not have a statistically significantly higher concentration of CGA than the noradrenergic phenotype. The CGA concentration of the adrenergic and noradrenegic phenotype is shown in Figure 2 as the notched box plot.

The clinical sensitivity of 71 PPGL patients based on plasma CGA values greater than 150 μg/L before surgery is equal to 90%. A total of seven patients (three not specified, one *RET*, two *VHL*, one *NF1*) had pre-surgery CGA values below 150 µg/L (Figure 1). Five of these patients had a tumor volume less than 20 mm^3^, and in one PCC patient diagnosed in 2002 with VHL gene mutation, recurrence was observed in 2003, 2005 and 2014. A kidney tumor was removed from him in 2014. The last patient was treated with vasodilating and antithrombotic drugs. Clinical sensitivity based on the concentrations of plasma MN and NMN was 97%.

Clinical specificity of PPGL patients based on all plasma CGA values during one week, six months, one year and two years after surgery less than 150 μg/L is equal to 99% (n = 98, CGA 73.2 ± 28.4 [29.7–187.6] µg/L). One result concerning the specificity of CGA 188 µg/L six months after surgery was a false positive. There was also a clinical specificity of 99% for PPGL patients determined by the concentration of metanephrines.

Results concerning the clinical specificity of CGA in the group of control patients without PCC or PGL are shown in Table 2. Some of these control patients had a diagnosis closely related to PPGL. This concerned 44 patients with adrenocortical adenoma or one patient with adrenocortical adenoma together with medullary thyroid carcinoma (MTC), follicular thyroid carcinoma (FTC) or papillary thyroid carcinoma (PTC), two patients with MEN2A syndrome or seven patients with MEN2A and MTC, three patients with MEN2B and MTC, or nine patients with merely somatic MTC. There was one patient with MTC and PTC and four patients with PTC, nine patients with hypertension, and five cases in which patients suffered from thyroid diseases (one goiter, one thyroiditis, three thyroid nodules). This concerned a total of 86 cases. One of these patients with hypertension labeled CRI in Figure 1 was not included in the control group because he suffered from chronic renal insufficiency (4th degree hypertensive nephrosclerosis), so Table 2 shows a total of 85 patients. The CGA clinical specificity of the control group is 92%; seven out of 85 had a CGA higher than 150 µg/L. All of these patients were diagnosed with adenoma of the adrenal gland, which was accompanied by two serious cases of cardiovascular disease and two cases of hypertension. MN and NMN values of these patients were inside the reference range. The clinical specificity of the control patients, based on the concentration of metanephrines, was 99%.

The mass of the operated PPGL tumors was determined in 59 of a total of 78 PPGL patients. Of these 59 patients, five patients who received PPI and one patient with CRI were excluded. The correlation between mass of PPGL tumors and corresponding CGA was calculated in 53 patients according to the following equation: Mass (g) = 16.2481 + 0.2073 × CGA(µg/L); n = 53, correlation *r* = 0.4490, significance level *p* = 0.0007.

The volume of the operated PPGL tumors was determined in 47 of a total of 78 PPGL patients. Of these 47 patients, six patients who received PPI and one patient with CRI were excluded. The correlation between volume of PPGL tumors and corresponding CGA was calculated in 40 patients according to the equation: Volume (mm^3^) = −23.6810 + 0.3131 × CGA (µg/L); n = 40, correlation *r* = 0.7300, significance level *p* = 0.0000.

The PASS score [29] was determined in 65 of a total of 78 PPGL patients (PASS score mean ± SD = 5 ± 3, range 1–11 points). Of these 65 patients, six patients who received PPI and one patient with CRI were excluded. The correlation between PASS scores and the corresponding CGA was calculated in 58 patients according to the equation:PASS = 3.7643 + 0.0023 × CGA (µg/L); n = 58, correlation *r* = 0.4322, significance level *p* = 0.0007.

## 3. Discussion

In a pilot study of 25 PPGL patients in 2008, we examined the utility of the CGA radioimmunoassay for the diagnosis of PPGL [30]. The results were so hopeful that by 2017 radioimmunoassays with CGA and plasma free metanephrines were determined in our institute in another 55 PPGL patients and results were published in 2017 [47]. This article includes all data on CGA and free plasma metanephrines concerning 78 PPGL patients measured by radioimmunoassays in our institute. Data were revised according to clinical diagnosis, consistently purified from all results of PPI-mediated analyses, chronic renal insufficiency or recurrence of the disease. Based on the revised results, we had to reduce clinical sensitivity for CGA from 93% [47] to 90%, and for metanephrines we increased the clinical sensitivity from 96% [47] to 97%. The combined clinical specificity calculated from the results of PPGL patients four months and more after surgery and from the control group (151 analyses) was 96% for CGA and 100% for metanephrines [47] and it was decreased in this set (156 analyses) to 95% for CGA and 99% for metanephrines. In this study the clinical specificity of PPGL patients after surgery (98 analyses over one week to two years after the operation of PPGL; analyses influenced by PPI, CRI or REC were excluded—see Figure 1) was 99% for CGA and 99% for metanephrines. The clinical specificity of the control group (85 analyses, one patient with CRI was excluded) was 92% for CGA, 99% for metanephrines. In all cases the combination of metanephrines and CGA gave 100% results of clinical sensitivity.

The clinical sensitivity of CGA in the PPGL group of patients is 90% and is therefore below the clinical sensitivity determined by MN and NMN (97%). But the sensitivity of metanephrines is not equal to 100%. In our PPGL patient population, metanephrines were falsely negative in one case of PGL and in one case of PCC (the kind of genetic mutation is unknown), while CGA levels were increased. In the PGL case the preoperative concentrations of CGA and NMN were increased, while MN was normal. The patient was designated for resection of sympathetic paraganglioma due to identical results of imaging procedures using MIBG and FDG PET, respectively. The PGL located on the abdominal aorta was removed during the first operation and one week after surgery the CGA concentration was increased, NMN and MN were within the normal reference range. Further measurements after 6 months have shown that the CGA level is still elevated, but MN and NMN are normal. By the imaging procedure (PET/CT with FDOPA) another deposit was found and PGL was surgically removed in the area of the right adrenal gland after 10 months. One week after surgery the concentrations of CGA, MN and NMN were within the normal reference range. Table 1 does not state that one PGL patient with recurrence of the disease had MN and NMN within the reference range in the first week after surgery, while the CGA was greater than the CGA cut-off value. Zuber et al. [26] showed that CGA is a valuable complementary biomarker in the workup of SDHB-related PCC/PGL. Combined with plasma NMN, CGA further enhance tumor detection by 22% with minimal loss in specificity. Unlike the previous quote, other literature [49] states that plasma CGA levels are increased in only a small portion (16%) of patients with biochemically silent hereditary head and neck paragangliomas, but the finding that nine out of 62 PGL patients with a biochemically silent tumor had an elevated CGA level possesses diagnostic significance. The CGA can be increased in a number of neuroendocrine neoplasias, so it cannot be used alone to diagnose PPGL. Metanephrines also have an advantage over CGA that they are unaffected by PPI use, and renal insufficiency should not affect their value. However, the combination of CGA and plasma metanephrines increases the predictive value in terms of clinical sensitivity and specificity and it is evident that CGA determination can be an appropriate addition to MN and NMN assays in laboratory diagnostics of PPGL patients.

The clinical specificity of PPGL patients after surgery based on all plasma CGA values (n = 98) purified from results with PPI, CRI or REC (Figure 1) is 99% and is the same as the 99% specificity based on the MN and NMN assays. CGA determination was also in this case an appropriate addition to the MN and NMN tests, and vice versa.

The clinical specificity of CGA in the control group was 92%, while the clinical specificity based on the MN and NMN tests was 99%. In 1 case of a patient with adrenocortical adenoma and MTC without diagnosis of PPGL, the NMN was falsely positive increased (197 ng/L), while the CGA 115 µg/L was below the 150 µg/L cut-off value.

A PASS score was absent in the nine patients without surgery and it was not determined in four patients. Of the 58 PPGL patients who had determined the PASS score, 46 patients (79%) had the PASS score equal to or greater than 4. In four metastatic PCC patients, PASS score was also equal to or greater than 4 (mean PASS ± SD = 7 ± 3). A PASS score equal or higher than 4 indicates uncertain biological behavior and increases the likelihood of malignancy. From this perspective, the PASS score was clinically relevant in malignant PPGL patients. We found in the literature [50] that the overall sensitivity for the PASS algorithm to correctly identify a malignant PCC (n = 105) was 97%, whereas the specificity of benign PCC (n = 704) was 68%. The sensitivity of the PASS score in malignant PGL (n = 13) was 100%, the specificity of benign PGL (n = 29) was 72%.

In the monitored PPGL and control group, increased values of CGA were found in connection with the PPGL diagnosis, while for patients with MEN syndrome and MTC or with differentiated cancer of thyroid gland (PTC, FTC) without the presence of PCC or PGL, CGA values were within the reference range.

Although PPGLs are rare tumors of chromaffin cells, it has serious consequences to the health of afflicted patients. Its familiar occurrence in individuals with a genetic predisposition is common in young people and often consists of bilateral tumors with aggressive biological behavior [7]. Unlike the determination of MN or NMN, the determination of CGA is not influenced by hypertensive drugs, substances and activities affecting biosynthesis and the secretion of catecholamines [7,12], but the value of CGA determination in patient taking proton pump inhibitors and in patients with severe renal insufficiency is limited [4,5] (see Figure 1). 

The Endocrine Society [7] recommends that initial screening for PPGLs should include measurement of plasma-free metanephrines or urine-fractionated metanephrines using liquid chromatography with mass spectrometric or electrochemical detection methods. The CGA in circulation seems to be in general a biomarker of neuroendocrine tumors which can improve diagnosis, serve to estimate prognosis and monitor the course of treatment [5]. CGA is an essential protein for PPGLs and if immunohistochemistry for CGA is negative, PPGLs should be ruled out [25]. According to the European Society of Endocrinology [51], CGA should be preoperatively measured in patients with normal preoperative plasma or urinary levels of MN, NMN and 3-methoxytyramine (3MT), and also 2–6 weeks and every year after surgery. Elevated preoperative CGA levels can be used to screen for local or metastatic recurrences or new tumors. The postoperative determination of CGA is recommended in cases with preoperative elevated CGA and normal metanephrines. Our experience corresponds to the recommendations of the European Endocrinology Society. We think that if there is any doubt about metanephrines in the laboratory diagnosis of PPGL, CGA should also be determined. 

A problem in the immunoanalytical determination of CGA lies in the fact that immunoassays are not standardized, and the results of various kit producers are different. This is due to the different specificity and sensitivity of the antibodies used, as well as to the different format of the given immunoassays [5]. CGA is highly acidic, it binds Ca^2+^ ions and aggregates rapidly. If the Ca^2+^ concentration is limited in EDTA plasma samples, there CGA does not form aggregates and concentrations of free CGA are immunoanalytically detected in comparison with serum samples [52]. The principal limitation of CGA evaluation is its low clinical specificity and sensitivity. The CGA is elevated in PPGLs, however, as well as in other neuroendocrine or non-neuroendocrine neoplasia and in clinical conditions with increased adrenergic activity [4]. The advantage of the immunoassay approach is that these methods are common in a clinical-biochemical laboratory and do not require additional expensive instrumentation. Limitations are given by the immunoassay principle in which, owing to interference, the immunological quantity does not have to correspond to the biological activity.

## 4. Materials and Methods

### 4.1. The Group of Patients

Included in the study were the group of PPGL patients and the control group without PPGL diagnosis. Patients with PCC or PGL consisted of 78 patients aged 49 ± 17 (18–78) years, 36 females aged 52 ± 18 (18–78) years, 42 males aged 48 ± 15 (20–76) years. PCC occurred in 68 patients, PGL in 10 patients. In the PCC group, mutations in the *RET* gene were found in seven patients, three were diagnosed with MEN2A and MTC, four had MEN2B and MTC. Mutations in the *VHL* gene were observed in five patients, in three patients were mutations in the *NF1* gene, two patients had mutations in the *MAX* gene, one patient in the *MET* gene, and in 50 patients the mutations were not determined. In the PGL group, mutations in the *SDHB* gene were found in two patients, while mutation in the *SDHD* gene was observed in one patient. Mutations were not determined in seven patients. 

A sampling of biological materials was performed preoperatively (2 ± 3 weeks before surgery) and about one week (1 ± 0.5 weeks), six months (30 ± 10 weeks), one year (54 ± 6 weeks) and two years (96 ± 22 weeks) after surgery. A blood basal sampling was performed 20 min after the introduction of cannula in supine position. Four collections (before surgery, one week, six months, one year after surgery) were conducted in 27 patients (25 PCC, 2 PGL); 19 blood samplings were from patients treated with proton pump inhibitors. Three collections (before surgery, one week, six months after surgery) were from seven PCC patients, in which four samplings were from patients treated with PPI and one patient suffered from chronic renal insufficiency (CRI). Three collections (before surgery, one week, one year after surgery) were from one PCC and one PGL patient, one sampling of PGL patient was treated with PPI. Two collections (before surgery, one week after surgery) were from six PCC patients and two PGL patients, three samplings were from patients treated with PPI, one PGL patient had a recurrence of the disease (REC). Two collections were in one patient (before surgery, six months after surgery), two collections (before surgery, one year after surgery) were from three PCC patients. Two collections were in 10 PCC patients and two PGL patients (before surgery, two years after surgery), two PCC patients had REC of the disease. One collection was in 15 PCC patients and three PGL patients before surgery.

The control group consisted of 86 patients aged 53 ± 19 (7–84) years, 57 females and 29 males without diagnosis of PCC or PGL. These patients, however, had adrenal problems, or were diagnosed with MEN syndrome and with medullary, papillary or follicular thyroid carcinoma. The rest of the control group suffered from thyroid diseases (goiter, thyroiditis, thyroid nodules). Forty six patients had adrenocortical adenoma, two of whom suffered from hypertension; two patients had serious cardiovascular problems, one adenoma was accompanied by MTC and FTC, one adenoma was accompanied by PTC. Nine patients suffered from hypertension, one of these patients had severe chronic renal insufficiency. Nine patients had MEN2A syndrome, which was associated with MTC in seven patients. MTC was detected in addition to MEN2B in three patients. Ten patients suffered from MTC, one of whom was also diagnosed with PTC. Four other patients were found to have PTC. Five patients had thyroid disease (one goiter, one thyroiditis, and three thyroid nodules). The study was approved by the Internal Grant Agency, Ministry of Health of the Czech Republic, grant No. NT/12336-4 and local Ethical Committee, which also examined and adopted the informed consent of the patient. 

### 4.2. Laboratory Examination

CGA was determined in the EDTA-plasma using a commercially available a solid-phase two-site immunoradiometric assay with primary immobilized monoclonal antibodies and secondary radioiodinated monoclonal antibodies, both directed against sterically remote sites on the CGA molecule (manufacturer Cisbio Bioassays, Codolet, France; code CGA-RIACT). The cut-off value of CGA in the EDTA plasma 150 µg/L was determined on the basis of our results and information from the Cisbio company that 97% of healthy persons (n = 60) had CGA below 97 µg/L with a maximum concentration of 146 µg/L. 

Free MN and NMN were determined in human EDTA plasma by competitive radioimmunoassay using a MetCombi Plasma RIA kit (IBL International GmbH, Hamburg, Germany, code RE29111). The cut-off values were 100 ng/L for metanephrine and 170 ng/L for normetanephrine.

### 4.3. Statistics

All statistical calculations were made using the computer program NCSS 2004 (Number Cruncher Statistical Systems, Kaysville, UT, USA). 

## 5. Conclusions

We can conclude that plasma CGA determined by immunoassay, which is simple without the necessity of special laboratory equipment, is an effective marker of PPGLs and can serve as an appropriate complement to MN and NMN assays in laboratory diagnosis of PPGL patients. Based on 90% clinical sensitivity, CGA can provide additional information to metanephrines, but absence of elevated CGA should not be relied on to rule out PPGL. In all cases of PPGL patients under investigation the combination of metanephrines and CGA gave 100% results of clinical sensitivity. Plasma CGA also exerts an association to PASS score, tumor mass and tumor volume.

## Figures and Tables

**Figure 1 cancers-11-00586-f001:**
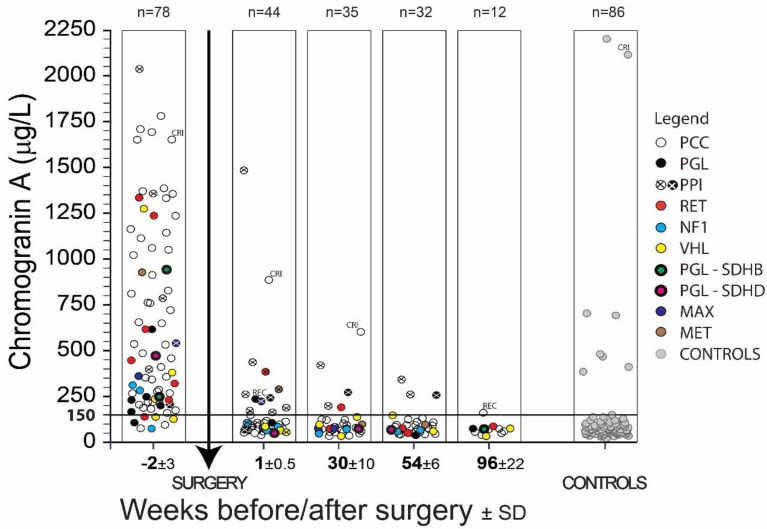
The results of chromogranin A (CGA) determination in PPGL patients before and after surgery. Symbols indicate the type of PPGL and found mutations in genes *RET*; *NF1*; *VHL*; *SDHB*; *SDHD*; *MAX*; *MET*. The meaning of abbreviations is pheochromocytoma (PCC); paraganglioma (PGL); patients treated with proton pump inhibitors (PPI); patients with chronic renal insufficiency (CRI); patients with a recurrence of the disease (REC).

**Figure 2 cancers-11-00586-f002:**
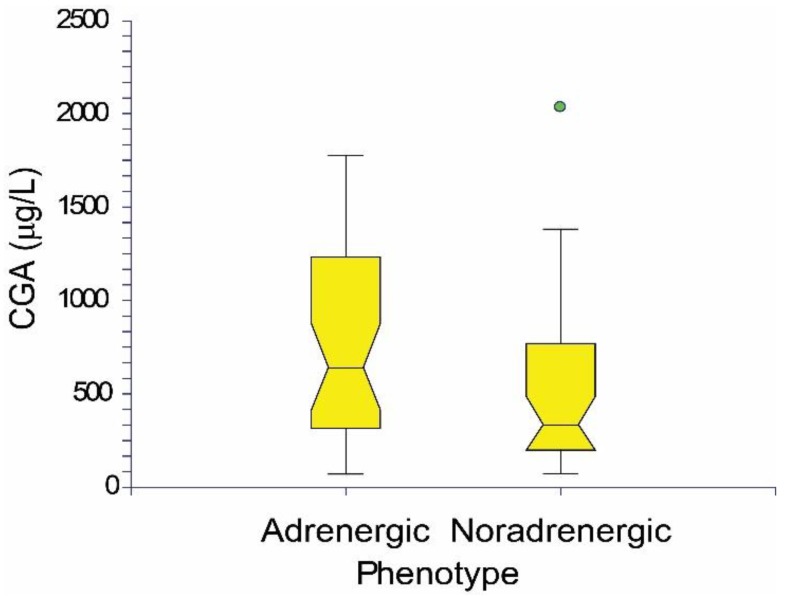
The notched box plot of CGA concentrations in adrenergic and noradrenergic phenotype of PPGL patients.

**Table 1 cancers-11-00586-t001:** The clinical sensitivity and specificity of PPGL patients based on the plasma chromogranin A (CGA) determinations before and after the adrenal surgery. The meaning of abbreviations is pheochromocytoma (PCC); paraganglioma (PGL); together denoted as PPGL; mutations in genes *RET*; *VHL*; *NF1*; *MAX*; *MET*; *SDHB* and *SDHD*; number (n); standard deviation (SD).

PPGL—Patients with Pheochromocytoma (PCC) or Paraganglioma (PGL)																								
Disease	Genetics	Gender	n	2 ± 3 Weeks before Surgery					1 ± 0.5 Week after Surgery				6 Months (30 ± 10 Weeks) after Surgery				1 Year (54 ± 6 Weeks) after Surgery				2 Years (96 ± 22 Weeks) after Surgery			
				AGE (Years)		CGA (µg/L)		Clinical Sensitivity	n	CGA (µg/L)		Clinical Specificity	n	CGA (µg/L)		Clinical Specificity	n	CGA (µg/L)		Clinical Specificity	n	CGA (µg/L)		Clinical Specificity
				Mean	SD	Mean	SD			Mean	SD			Mean	SD			mean	SD			Mean	SD	
**PPGL**	**total**	total	71	49	17	618.9	479.7	90	27	76	24.4	100	31	75.9	33.8	97	29	73.1	28.6	100	11	59.2	16.7	100
		males	38	47	16	638.7	512	92	15	69.1	21.5	100	17	71.6	28.6	100	16	73.1	31	100	6	64.1	16.6	100
		females	33	50	18	596.1	446.5	88	12	84.6	26	100	14	81.2	39.7	93	13	73.2	26.6	100	5	53.4	16.7	100
**PCC**	**total**	total	62	48	17	657.1	492.7	90	25	76	24	100	30	76.2	34.3	97	27	74.9	28.6	100	9	57.0	17.9	100
		males	35	48	16	667.4	522.8	91	14	70.7	21.3	100	16	71.8	29.5	100	15	73.7	32	100	4	61.5	20.8	100
		females	27	48	18	643.7	460.4	89	11	82.8	26.5	100	14	81.2	39.7	93	12	76.4	25	100	5	53.4	16.7	100
	**not specified**	total	45	52	16	718.9	506	96	20	77.1	26.4	100	20	71.2	27.3	100	20	74.8	27.5	100	6	55.0	13.6	100
		males	25	52	15	758.4	543.4	100	10	70.6	24.6	100	11	70.7	24.8	100	10	70	30	100	3	54.3	18.2	100
		females	20	52	16	669.6	464.2	90	10	83.7	27.8	100	9	71.8	31.5	100	10	79.6	25.5	100	3	55.7	11.4	100
	***RET***	total	7	34	9	614.8	482.6	86	-	-	-	-	2	129	82.8	50	2	60.5	20	100	1	83.3	-	100
		males	3	42	6	693.7	603	67	-	-	-	-	-	-	-	-	-	-	-	-	1	83.3	-	100
		females	4	28	5	555.6	460.9	100	-	-	-	-	2	129	82.8	50	2	60.5	20	100	-	-	-	-
	***VHL***	total	5	32	21	427.9	482.5	60	2	70.9	14.9	100	3	85	52.2	100	2	99.1	62.6	100	2	49.9	28.5	100
		males	3	22	2	244.2	127.4	67	2	70.9	14.9	100	2	82.3	73.5	100	2	99.1	62.6	100	-	-	-	-
		females	2	46	34	703.6	803.5	50	-	-	-	100	1	90.3	-	100	-	-	-	-	2	49.9	28.5	100
	***NF1***	total	3	54	10	220.2	128.8	67	2	71.4	15.4	100	2	56.2	16.9	100	2	57.8	7.5	100	-	-	-	-
		males	3	54	10	220.2	128.8	67	2	71.4	15.4	100	2	56.2	16.9	100	2	57.8	7.5	100	-	-	-	-
		females	-	-	-	-	-	-	-	-	-	-	-	-	-	-	-	-	-	-	-	-	-	-
	***MAX***	total	1	60	-	357.9	-	100	1	74.2	-	100	2	71.5	2.1	100	-	-	-	-	-	-	-	-
		males	-	-	-	-	-	-	-	-	-	-	-	-	-	-	-	-	-	-	-	-	-	-
		females	1	60	-	357.9	-	100	1	74.2	-	100	2	71.5	2.1	100	-	-	-	-	-	-	-	-
	***MET***	total	1	47	-	925	-	100	-	-	-	-	1	94.4	-	100	1	92	-	100	-	-	-	-
		males	1	47	-	925	-	100	-	-	-	-	1	94.4	-	100	1	92	-	100	-	-	-	-
		females	-	-	-	-	-	-	-	-	-	-	-	-	-	-	-	-	-	-	-	-	-	-
**PGL**	**total**	total	9	52	20	355.9	270.8	89	2	74.9	40.4	100	1	67.3	-	100	2	49.3	20.7	100	2	69.1	2.8	100
		males	3	36	12	303.7	145.1	100	1	46.4	-	100	1	67.3	-	100	1	64	-	100	2	69.1	2.8	100
		females	6	61	18	382	326.3	83	1	103.5	-	100	-	-	-	-	1	34.6	-	100	-	-	-	-
	**not specified**	total	6	56	22	258.2	181.1	83	1	103.5	-	100	-	-	-	-	1	34.6	-	100	1	71.1	-	100
		males	1	32	-	197.1	-	100	-	-	-	-	-	-	-	-	-	-	-	-	1	71.1	-	100
		females	5	61	20	270.5	199.7	80	1	103.5	-	100	-	-	-	-	1	34.6	-	100	-	-	-	-
	***SDHB***	total	2	55	7	592.2	190.9	100	-	-	-	-	-	-	-	-	-	-	-	-	1	67.1	-	100
		males	1	50	-	245.1	-	100	-	-	-	-	-	-	-	-	-	-	-	-	1	67.1	-	100
		females	1	60	-	939.4	-	100	-	-	-	-	-	-	-	-	-	-	-	-	-	-	-	-
	***SDHD***	total	1	26	-	469	-	100	1	46.4	-	100	1	67.3	-	100	1	64	-	100	-	-	-	-
		males	1	26	-	469	-	100	1	46.4	-	100	1	67.3	-	100	1	64	-	100	-	-	-	-
		females	-	-	-	-	-	-	-	-	-	-	-	-	-	-	-	-	-	-	-	-	-	-

**Table 2 cancers-11-00586-t002:** The clinical specificity determined from the control group of patients with various adrenal disorders; but not afflicted by pheochromocytoma or paraganglioma. The meaning of abbreviations is plasma chromogranin A (CGA); multiple endocrine neoplasia (MEN) type 2A (MEN2A) or 2B (MEN2B); medullary (MTC); papillary (PTC) or follicular (FTC) thyroid carcinoma; number (n); standard deviation (SD).

Controls							
Syndrome	Gender	n	AGE (years)		CGA µg/L)		Clinical Specificity
			Mean	SD	Mean	SD	
total	total	85	53	19	125.3	259.3	92
	males	29	53	21	102.6	128.9	93
	females	56	54	18	134	306.3	91
adrenocortical adenoma	total	44	63	11	178.5	352.7	84
	males	15	65	6	135.2	173.3	87
	females	29	62	13	200.9	417.7	83
adrenocortical adenoma + MTC, FTC	total	1	71	-	31.8	-	100
	males	-	-	-	-	-	-
	females	1	71	-	31.8	-	100
adrenocortical adenoma + PTC	total	1	75	-	108.4	-	100
	males	-	-	-	-	-	-
	females	1	75	-	108.4	-	100
hypertension	total	8	45	16	43.2	11	100
	males	4	41	22	50.2	9.1	100
	females	4	48	8	36.3	8.3	100
MEN 2A	total	2	10	4	70.6	15.2	100
	males	1	13	-	81.3	-	100
	females	1	7	-	59.8	-	100
MEN 2A + MTC	total	7	26	14	69.5	35.5	100
	males	3	20	12	71.8	24.9	100
	females	4	30	16	67.7	45.7	100
MEN 2B + MTC	total	3	30	16	90.7	53.9	100
	males	1	17	-	146.2	-	100
	females	2	37	16	62.9	34.4	100
MTC	total	9	56	13	76.9	36.2	100
	males	2	51	17	51.3	10.9	100
	females	7	57	13	84.2	38.1	100
MTC + PTC	total	1	70	-	42.3	-	100
	males	1	70	-	42.3	-	100
	females	-	-	-	-	-	-
PTC	total	4	58	22	81.1	40.4	100
	males	1	80	-	102.6	-	100
	females	3	50	20	73.9	46.2	100
thyroid disorders	total	5	36	18	70.2	12.2	100
	males	1	51	-	54.6	-	100
	females	4	32	18	74.1	9.9	100

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
