# Peer review of "Chromogranin A in the Laboratory Diagnosis of Pheochromocytoma and Paraganglioma"

_cancers, 2019, doi:10.3390/cancers11040586_

Round 1
Reviewer 1 Report
The manuscript has clearly improved from the previous version. However, the value of sole determination of CGA to rule out recurrence remains of limited clinical value, due to the fact, that the sensitivity of CGA was merely 90% for PPGL. Similarly to pre-OP, 1 out of 10 patients with PPGL will be missed. For that reason the conclusion of the paper needs to clearly indicate that CGA can provide additional information to metanephrines, but absence of elevated CGA should not be relied on to rule out PPGL. As the authors correctly state, testing CGA can be useful in patients where metanephrines are not elevated, either for diagnosis or follow-up (when CGA was elevated at initial diagnosis).
Publication of the author's experience appears appropriate for publication. Below are some suggestions to further improve the article.
1) Clearly stating the research/clinical question which the authors aim to answer in the abstract and at the end of the introduction will help the reader to follow the argument of the manuscript.
2) An introductory sentence for the results section stating what has been measured in which patients will improve readability and interest to the reader. Jumping right into the figure legend may appear confusing.
3) Please state how recurrence of disease or metastases have been determined. Based on metanephrines and/or imaging? It should be clarified, for how many of the post-OP patients this has been performed. Given a sensitivity of 90% for detection of PPGL by elevated CGA, more sensitive methods to confirm absence of disease at follow up should be taken and provided with the presented data. How was recurrence ruled out in the patient with elevated CGA and RET mutation?
4) Apparently the evaluated cohort included two patients with normal metanephrines but elevated CGA. Relevant disease characteristics (PPGL location, absence of SDHB immunohistochemisty, adrenergic or noradrenergic phenotype, signs and symptoms that led to suspicion of PPGL) of these patients will be of interest and should be mentioned. Detection of recurrence/residual disease in on of these patients is clearly of interest and clinical importance.
5) The authors state that recurrence (a separate PPGL mass) developed within 1 week post-OP. This appears highly unlikely, since PPGLs are slow growing tumors. Oversight of this separate lesion previous to surgery is much more likely and should be discussed. Similarly to determination of metanephrines, CGA may be useful in such cases to determine residual disease post-OP. Especially in PPGLs without elevated metanephrines, this is of high importance.
6) Please clarify how the cut off for plasma EDTA was determined. In the methods section, it sounds like the highest measured level from a control person indicated by the developer was used? If that was the case, please discuss why that appeared appropriate.
7) Please state why the control group is relevant to determine the specificity of a PPGL marker. Was there any reason to suspect PPGL in these patients? The authors state, that hypertension can be a reason for elevated CGA. Thus, similar to the patients with adenocortical adenomas, a control cohort of hypertensive patients will most likely lead to a much higher rate of false positive CGA levels. This should be discussed.
8) p6 l 193: replace malignant with metastatic. Please elaborate on these patients. Did the 4 patients surviving past surgery have elevated CGA after removal of the primary tumor? Did metastases alone produce elevated CGA levels? In the introduction the authors state, that CGA is lost in dedifferentiating, aggressive NETs. Is that also evident for any of their metastatic PPGL?
9) The comparison of adrenergic and noradrenergic phenotype with CGA appears interesting and may justify an additional figure
10) It remains unclear whether head and neck paragangliomas have been included. If that was not the case, that should be indicated in the material and methods section.
11) p7 l 207: the years 2003, 2005, 2014 should be put in relation to the year of initial diagnosis of the patient. Without that, the mention of the year is of not value to the reader.
12) Figure 1 legend states that data for patients with adrenal surgery is depicted. That does not include PPGL cases. Please clarify
13) It remains unclear how the correlation of CGA with PASS is relevant. Grouping patients with metastatic disease or recurrence and those without any detected recurrence at the last follow-up may be of interest here. Of note: PASS is a score developed for adrenal pheochromocytomas, thus a comparison of adrenal and extra-adrenal PPGLs may also be of interest (even the higher risk for metastatic disease in the latter). Furthermore, elevated CGA with elevated PASS may also be related to the extent of disease (more related to tumor mass/volume). Thus, anaysis to determine a correlation between tumor volume and PASS score also appears of interest.
14) In the discussion the authors state that they had to correct their previously estimated specificity for metanpehrines for detection of PPGL from 100% to 99% in the same patient group. What was the reason? Did the authors rule out medication or nutritional reasons leading to elevated metanephrines in the patient without PPGL (caffeine, paracetamol, antidepressants etc.)? This should be discussed. Perhaps there was a second metanephrine determination after exclusion of interfering substances? Furhermore, the decreased sensitivity and specificity of immunoassays compared to mass spectrometry based determination of metanephrines should be discussed here.
15) P9, l277: bearing? replace with located?
P9, l280: usually the own results are referred to in the past tense
P9, l283: correct: within the normal reference range
l287: further enhance
l291: neoplasias
l294: can be an appropriate addition
l316: it has serious consequences to the health of...
l317: individuals with a genetic predisposition is common in your people and often consists of bilateral tumors with aggressive ...
l320: citation needed for interference of hypertensive drugs with metanephrine values
l321: ..., but the value of CGA determination in patients taking proton pump inhibitors....
p10 l323: ... initial screenig for PPGLs should include measurement of ...
p10 l327: it should be specified that the cited article refers to neuroendocrine tumor in general or specified which aspect of the article is referred to.
p10 l330: It should be specified that post-OP CGA is recommended in cases with pre-OP elevated CGA and normal metanephrines, not in general.
l338: low? affinity? is this correct?
l339:..., there CGA does not form aggregates and concentrations of free CGA...
16) The discussion should include that contrary to CGA, metanephrines can also be utilized in patients with renal insufficiency and in patients taking H2 blockers and thus would be useful in the 6 patients excluded pre-OP and 14 non evaluable post-OP patients, further arguing towards always measuring metanephrines.
17) The conclusion section should be following directly after the discussion. I disagree with the conclusion, that CGA is an effective marker of PPGLs. At 90% sensitivity, 1 in 10 patients is missed. Under current clinical standards, that appears a rate much too high to consider effective diagnosis.
Reviewer 2 Report
The revised manuscript is improved, but numerous grammatical errors as well as fact errors persist. Some of the typos and grammatical inconsistencies makes it hard to follow the results. For example:
1. Rows 68-69:
"PPGLs have considerable genetic heterogeneity [15] and they are divided into different clusters with similar pathogenesis and biology."
This is not correct. The main reason we discuss PPGLs from a clustering standpoint is the association of various clusters to different patient outcomes and underlying genetics. The pathogenesis and biology is therefore NOT similar.
2. Row 130:
"The PASS score is in most cases above four [28]."
Yes, in the referred article perhaps, but the phrasing is a dangling modifier and makes it sound like any given PASS score regularly is >4 - which is incorrect.
3. Row 193:
"Six of the PCC patients (none of the PGL patients) were malignant"
Tumors are malignant, not patients.
4. Rows 304-205:
"Fifty eight PPGL patients with a known PASS score of 46 (79 %) exhibited a PASS score equal to or higher than 4, including all 4 cases with malignant behavior (PASS score mean ± SD = 7 ± 3)."
This phrasing makes it sound like 58 patients had a PASS score of 46. Please correct.
Author Response
Please see attached file.

This manuscript is a resubmission of an earlier submission. The following is a list of the peer review reports and author responses from that submission.
Round 1
Reviewer 1 Report
Bilek et al. resubmitted an article, arguing for the utility of chromogranin A determination in plasma samples by radioimmunoassay to correctly classify paraganglioma patients. The authors have responded sloppily to the comments and the problems have not really been ruled out. In addition, the authors confirm 100% overlap with previously published data. The main difference appears to be reanalysis after some final diagnoses had been revised from the original publications. In my opinion this should be performed in form of an erratum to the original article and is not sufficient for novel publication. The authors state data on chromogranin and metanephrines is presented in the current paper, however, no metanephrine data is presented.
For that reason I recommend the article again for rejection by the high ranking journal Cancers.
Reviewer 2 Report
The revised version is improved compared to the original submission, but some important points remain to consider:
1. The authors are now much more transparent regarding their earlier work performed on this cohort, at least in the Discussion section. The authors also need to clarify that this work is only partly original in the Introduction section as well.
2. The association between earlier PASS scores (from a previous publication) and CGA levels is an interesting addition to the revised version, but is not mentioned in the Discussion at all - but reappears in the Conclusions section. I think this finding should be briefly discussed, as it is potentially clinically relevant - not least the recent coupling between Chromogranin B levels and PASS scores in PPGL. Why weren't PASS scores from all PPGLs included in his analysis? This should be mentioned.
3. The newly added text in the first paragraphs of the Discussion section needs extensive English grammar revision - it is very hard to follow.
4. Rows 235-239: (...after surgery (98 analysis)) et.c. What does "98 analysis" mean? There are several phrases like this across this section which needs revision and clarification.
Reviewer 3 Report
In general, the manuscript of Bilek and coworker has been improved during the first iteration. The language still needs to be fixed. The manuscript mainly summarized two already published studies. By this retrospective approach and the resulting increase in patient number, the clinical specificity and sensitivity of CGA could be corrected.
Comments:
Abstract: In the results and discussion section, you summarize data from 78 patients, in the abstract only of 71. Please correct.
Again, please read the literature for HIF2a or EPAS1 (https://www.uniprot.org/uniprot/?query=EPAS1&sort=score). The gene have two different names!
Page2, line 69: Better: PPGLs are divided into different cluster…
Question: Are there differences between adrenergic and noradrenergic PPGLs in CGA secretion? Answer: CGA (noradrenergic) = 1786 ± 2535 g/l is statistically significantly higher than CGA (adrenergic) = 522 ± 563 µg/l, Please add the information in the text. It’s quite important.
Line 255: E.g. literature… Rewritten the sentence. The group of … stated that…
Line 276: Although PPGLs are rare tumors… Plural without a
Line 289: delete no
Line 357: replace material with equipment and are with is